# Mismatch Between Perceived and Actual Dietary Nutrition in Hospitalized Cardiovascular Patients and Clinicians: A Cross-Sectional Assessment and Recommendations for Improvement

**DOI:** 10.3390/nu17162624

**Published:** 2025-08-13

**Authors:** Di Li, Jiaheng Han, Ye Peng, Xi Yu, Ying Xiao, Junxian Song, Peng Liu

**Affiliations:** 1Department of Clinical Nutrition, Peking University People’s Hospital, Beijing 100044, China; lidi@pku.edu.cn; 2Faculty of Medicine, Macau University of Science and Technology, Taipa, Macao SAR 999078, China; 1210022651@student.must.edu.mo (J.H.); pengye@must.edu.mo (Y.P.); xyu@must.edu.mo (X.Y.); yxiao@must.edu.mo (Y.X.); 3Department of Cardiovascular Medicine, Peking University People’s Hospital, Beijing 100044, China

**Keywords:** cardiovascular disease, hospitalized patients, dietary awareness, clinical collaboration

## Abstract

**Background:** Multiple studies demonstrated that nutritional risk and malnutrition were associated with prolonged hospitalization, extended rehabilitation duration, and increased mortality among patients with cardiovascular diseases (CVD). However, current research on dietary behaviors and nutritional status in hospitalized CVD patients remains insufficient. **Objective:** This study systematically evaluated the concordance between cardiology inpatients’ and clinicians’ subjective nutritional status assessments and objective energy and protein intake achievement rates, while comprehensively investigating the multidimensional associations among Nutritional Risk Screening 2002 (NRS 2002), Global Leadership Initiative on Malnutrition (GLIM), blood parameters, and dietary intake. **Methods:** This study adopted a cross-sectional design to investigate hospitalized patients in the department of cardiology. Dietary knowledge and behavior data were collected through questionnaires, and actual dietary intake was recorded. Nutritional risk assessment and malnutrition diagnosis were performed for all inpatients. Differences between subjective evaluations and actual intake were compared, and the correlation between blood biochemical indicators and nutritional status was analyzed. **Results:** The study enrolled 618 valid cases, with male and female patients accounting for 67.48% and 32.52%, respectively. The patients’ age was 61.89 ± 12.88 years. The NRS 2002 score was 3.01 ± 0.94, with 132 inpatients diagnosed with malnutrition according to GLIM criteria. Energy and protein intake reached only 63.09 ± 18.23% and 74.98 ± 22.86% of target values, respectively. NRS 2002 showed significant correlations with estimated glomerular filtration rate (eGFR), C-reactive protein (CRP), albumin (ALB), etc. No significant difference was found between physician and inpatient evaluations (*χ*^2^ = 1.465, *p* < 0.05). Both ordinal and multivariable logistic regression analyses demonstrated significant discrepancies between subjective assessments (inpatient perceptions and physician evaluations) and objective energy and protein intake levels (*p* < 0.05). **Conclusions:** Hospitalized cardiovascular patients commonly exhibited insufficient nutritional intake and limited dietary awareness. A mismatch existed between patient/clinician perceptions and objectively assessed nutritional intake. Subjective evaluations could not accurately reflect actual nutritional status, necessitating enhanced nutritional monitoring—including nutritional risk screening, biochemical testing, and dietary surveys—along with personalized interventions. Future efforts should enhance collaboration between clinicians and dietitians to improve patients’ nutritional status and clinical prognosis.

## 1. Introduction

Cardiovascular diseases (CVD), comprising coronary artery disease (CAD), cerebrovascular disease, peripheral arterial disease (PAD), and aortic atherosclerosis, represents the leading chronic non-communicable diseases threatening global public health. As a pathological condition characterized by high morbidity and mortality, CVD remains one of the primary causes of death worldwide, with mortality rates surpassing the combined fatalities from all cancer types and chronic lower respiratory diseases [1,2]. Epidemiological data indicated approximately 330 million CVD patients in China, including 245 million hypertension cases and 13 million stroke survivors. Globally, CVD accounted for an estimated 17.3 million annual deaths, with projections suggesting this figure would rise to 23 million by 2030 [3]. World Health Organization (WHO) estimates suggested that 85% of cardiovascular mortality originated from myocardial infarction and stroke events, imposing substantial burdens on healthcare systems worldwide [4]. This challenge has been exacerbated by rapidly aging populations, creating pressing demands for preventive interventions. Emerging evidence has identified nutritional status as a critical modifiable factor, with studies demonstrating associations between malnutrition and cardiovascular complications [5]. Nutrition education has been shown to effectively improve dietary behaviors and nutrient intake [6,7,8]. As Jørgensen et al. demonstrated, modification of risk factors (e.g., unhealthy lifestyles) could prevent up to 80% of CVD cases [9]. Research revealed that low intake of protective dietary components coupled with excessive consumption of detrimental elements significantly increased CVD mortality [10]. The Mediterranean diet and Dietary Approaches to Stop Hypertension (DASH) diet demonstrate significant preventive effects against CVD incidence and prevalence reduction in clinical and epidemiological studies [11,12,13]. Consequently, medical societies and associations in China and internationally have incorporated dietary interventions and lifestyle modifications into both primary and secondary CVD prevention guidelines, as well as cardiac rehabilitation protocols.

Malnutrition is defined as a state resulting from inadequate, excessive, or imbalanced intake of energy and/or nutrients [14,15]. The study revealed that 57% of heart failure inpatients were at nutritional risk [16]. Among overweight/obese CVD inpatients, 42.4% were at risk of malnutrition, while 7.1% met diagnostic criteria for malnutrition [17]. Furthermore, malnutrition is a common complication in patients with CAD and is strongly associated with increased mortality [18]. The NRS 2002 score significantly influenced total hospitalization length and rehabilitation duration [19]. Patients with nutritional risk exhibited higher risks of complications and mortality compared to those without risk, with NRS 2002 scores ≥ 3 associated with over twofold increased all-cause mortality [20,21]. Malnutrition defined by GLIM criteria correlated significantly with impaired physical function and all-cause mortality. The prevalence of malnutrition in CVD patients was 18.9%, with particularly high rates among those aged > 60 years [22]. Malnutrition severity was linked to worsened CAD symptoms [23]. Patients with moderate/severe malnutrition faced elevated mortality risks versus non-malnourished individuals [24]. These findings underscore the critical role of dietary nutrition in CVD management. Prior studies identified inadequate nutritional knowledge and poor dietary habits in cardiac patients [25,26]. Harkin et al. reported that physicians generally possessed limited nutritional expertise, with only 13.5% feeling adequately trained to discuss nutrition with patients, which might lead to potential misjudgments of nutritional status by both patients and clinicians [27,28,29]. Data on nutrition support rates in cardiovascular inpatients remain scarce, but both patients and physicians demonstrated insufficient awareness of nutrition’s importance and suboptimal dietary cognition. However, research focusing on dietary knowledge and nutritional status among CVD patients remained limited, particularly those integrating patients’ dietary self-perceptions with physicians’ evaluations to determine the consistency between subjective dietary evaluations and objective nutritional status.

This study investigated dietary awareness, nutritional intake, and nutritional status among hospitalized cardiology patients in China. We hypothesized that subjective perceptions of nutritional status would discord with both actual dietary intake and objective nutritional parameters. Building upon this premise, the research focused on analyzing their dietary knowledge levels, behavioral patterns, and underlying determinants of malnutrition. The research evaluated whether physicians’ clinical evaluations and patients’ self-reported perceptions accurately reflected actual energy and protein intake levels (i.e., the consistency between perceived and objective nutritional intake), while identifying potential adverse factors affecting inpatients’ nutritional status.

## 2. Materials and Methods

### 2.1. Survey Group

This study conducted a three month cross-sectional survey at Peking University People’s Hospital. Using a random sampling method, we enrolled inpatients from the Department of Cardiovascular Medicine who met the following inclusion criteria: (a) age ≥ 18 years, (b) confirmed diagnosis of CVD, and (c) hospitalization duration ≥ 48 h. Exclusion criteria included the following: (a) cognitive impairment or (b) refusal to participate. Data were collected through face-to-face questionnaires combined with medical record reviews.

### 2.2. Survey Design

A self-developed questionnaire, titled “Survey on Dietary Nutrition Knowledge and Behaviors among Inpatients in the Department of Cardiovascular Medicine,” was used in this study. The first section, completed by the researchers, included patients’ general information (e.g., height, weight, age, and sex) and a comprehensive nutritional assessment conducted by dietitians, encompassing the Nutritional Risk Screening 2002 (NRS 2002) and malnutrition diagnosis based on the Global Leadership Initiative on Malnutrition (GLIM) criteria. The second section was administered via researcher-conducted interviews and consisted of 17 questions along with a detailed 24 h dietary recall (covering breakfast, morning snack, lunch, afternoon snack, dinner, and evening snack) to evaluate patients’ dietary knowledge and habitual eating behaviors. The third section involved an assessment of the inpatient’s dietary status by their attending physicians, rated on a 5-point scale (very good, good, general, bad, very bad).

### 2.3. Nutritional Risk Screening

Nutritional risk screening was performed for all patients using the NRS 2002 tool within 24 h of admission. The evaluators received standardized training prior to the study to ensure assessment accuracy and consistency. The screening process, as detailed in Appendix A, followed a two-stage protocol: initial screening assessed four key parameters (BMI, recent weight loss, dietary intake changes, and disease severity), with any positive finding prompting progression to the second stage involving three scoring domains (disease severity score, impaired nutritional status score, and age score) [30]. Patients scoring ≥ 3 were classified as nutritionally at-risk, prompting development of personalized nutritional support plans and intervention protocols. Those scoring < 3, while not requiring immediate nutritional therapy, were scheduled for weekly nutritional reassessment with close clinical monitoring.

### 2.4. Global Leadership Initiative on Malnutrition Criteria

The GLIM criteria represent an international consensus standard for diagnosing malnutrition. The diagnostic framework comprises two components: phenotypic criteria and etiologic criteria. According to the GLIM criteria, as shown in Appendix A, the diagnosis of malnutrition requires meeting at least one phenotypic criterion and one etiologic criterion [31]. Phenotypic criteria primarily reflect the physical manifestations of malnutrition, including involuntary weight loss, low BMI, and reduced muscle mass. The criteria for weight loss are as follows: 5–10% weight loss within the past 6 months or 10–20% weight loss over more than 6 months was classified as moderate; >10% weight loss within the past 6 months or > 20% weight loss over more than 6 months was classified as severe. The criteria for low BMI are as follows: BMI < 20 kg/m^2^ (<70 years) or <22 kg/m^2^ (≥70 years) was classified as moderate; BMI < 18.5 kg/m^2^ (<70 years) or <20 kg/m^2^ (≥70 years) was classified as severe. Reduced muscle mass had to be confirmed through clinical examination or body composition analysis, with mild to moderate reduction classified as moderate and severe reduction classified as severe. However, due to limitations in the cardiology ward, including the lack of instruments for precise muscle mass measurement, this study did not consider this phenotypic criterion. Etiologic criteria primarily reflect the underlying causes of malnutrition, including disease burden or inflammation, as well as reduced food intake or malabsorption. Based on the severity of the phenotypic criteria, GLIM classifies malnutrition into two grades: moderate (grade 1) and severe (grade 2).

### 2.5. 24 h Dietary Recall

Researchers visited the wards daily to record the 24 h dietary intake of newly admitted patients. For patients consuming hospital meals, precise food weight measurements were obtained from hospital kitchen records and cross-verified through post-meal observations. For those with external food sources, participants were instructed to photograph their meals before and after consumption. The actual intake of energy and protein was calculated based on the *China Food Composition Table* [32]. The target intake of energy and protein was estimated based on the actual situation and physical activity level of the patients, combined with the Chinese Society of Parenteral and Enteral Nutrition (CSPEN) and European Society for Clinical Nutrition and Metabolism (ESPEN) guidelines for enteral and parenteral nutrition, as presented in Table 1 [33,34,35].

### 2.6. Blood Parameters Collection

Blood parameters were recorded by review of the medical record system. The record included parameters related to blood lipids (total cholesterol (TC); triglycerides (TG); high-density lipoprotein (HDL); low-density lipoprotein (LDL)), liver function (alanine aminotransferase (ALT); aspartate aminotransferase (AST)), renal function (blood urea nitrogen (BUN); creatinine (Cr); estimated glomerular filtration rate (eGFR)), glucose metabolism (glucose (Glu); glycated hemoglobin (HbA1c)), inflammation (C-reactive protein (CRP)), nutrition (albumin (ALB); hemoglobin (Hb)), electrolytes (sodium (Na); potassium (K); calcium (Ca); inorganic phosphate (IP)), and uric acid (UA). These parameters were used to comprehensively assess the physiological and metabolic status of inpatients with CVD.

### 2.7. Statistics

Statistical analysis was performed using SPSS 27.0 software for data entry, management, and analysis to ensure data integrity and accuracy. In cross-sectional studies or survey-type research, the sample size was calculated using the following formula:$$Ν\;=\frac{{{Ζ}^{2}}_{\alpha /2}*Ρ*(1-P)*D}{{Ε}^{2}}$$

*P* represented the prevalence of inadequate dietary intake among hospitalized patients, which was set at 50% based on previous studies [36,37,38]. The severe malnutrition rate was defined as 5% for CVD patients [18,39]. *E* denoted the desired precision of the estimate, typically defined as 0.05. *Z_α_*_/2_ corresponded to the standard normal deviate (1.96 for a two-tailed test at *α* = 0.05). The design effect *D* was set to 1 for simple random sampling design [40]. Consequently, the calculated sample size was 384, and the sample size for severe malnutrition was 72 cases. Continuous variables were presented as mean ± standard deviation (*M* ± *SD*). Categorical data were expressed as counts and percentages [*n* (%)]. *T*-tests were used to compare differences between actual energy/protein intake and target requirements. Reliability analysis was conducted to assess the internal consistency of the objective items in the self-developed questionnaire, which included both 2-point and 5-point Likert scales. The chi-square test was employed to evaluate the consistency between physicians’ and patients’ subjective evaluations. Pearson’s linear correlation analysis was conducted to examine relationships between NRS 2002 nutritional risk scores and key blood parameters. Ordinal logistic regression modeling was applied to analyze significant blood parameters affecting malnutrition diagnosis and severity according to GLIM criteria. Ordinal logistic regression models were employed to assess the agreement between categorized energy and protein intake achievement rates and subjective evaluations, including both physician evaluations and inpatient perceptions. Multivariable logistic regression analysis was performed to identify the primary factors influencing both patient self-perceptions and physician evaluations of nutritional status. A two-tailed *p*-value < 0.05 was considered statistically significant for all analyses.

## 3. Results

### 3.1. Baseline Characteristics

The study initially enrolled 631 inpatients with CVD. After excluding 13 ineligible patients, a total of 618 valid cases were included. Among them, 417 patients (67.48%) were male, and 201 patients (32.52%) were female. The age of the patients was 61.89 ± 12.88 years, and the BMI was 25.49 ± 3.71 kg/m^2^.

### 3.2. Nutritional Risk and Malnutrition Prevalence

The NRS 2002 score was 3.01 ± 0.94, and 71.84% of the patients were at nutritional risk (NRS 2002 score ≥ 3). According to the GLIM criteria, 132 patients (21.36%) were diagnosed with malnutrition, including 114 cases of moderate (grade 1, 18.45%) malnutrition and 18 cases of severe (grade 2, 2.91%) malnutrition.

### 3.3. Dietary Survey

Figure 1A presents the dietary survey results, revealing that the mean actual energy intake of inpatients was 1117.03 ± 304.16 kcal/d, significantly lower than the target intake of 1801.94 ± 326.33 kcal/d (*p* < 0.05). Similarly, the mean actual protein intake was 48.03 ± 13.22 g/d, significantly lower than the target intake of 65.87 ± 13.97 g/d (*p* < 0.05), as shown in Figure 1B. On average, participants achieved 63.09 ± 18.23% of their target energy intake and 74.98 ± 22.86% of their target protein intake. The study found that 54.36% of patients achieved ≥ 60% of their target energy intake, while 14.08% reached ≥ 80% of the energy target. For protein intake, 79.21% of patients met ≥ 60% of the target, and 32.04% achieved ≥ 80% of the target.

### 3.4. Dietary Awareness Among Cardiovascular Inpatients

The validated questionnaire was administered to assess dietary knowledge and behavioral patterns among hospitalized patients with CVD, as shown in Appendix A. The Cronbach’s alpha coefficients for the objective items in the questionnaire were 0.598 (2-point scale) and 0.700 (5-point scale), respectively. 32.52% of patients expressed confusion regarding scientific dietary practices, while 56.31% reported having received nutritional education. Primary information sources included networks and newspapers, whereas only 3.40% consulted dieticians. Although most patients acknowledged the role of diet in disease management, 82.04% had not heard of the *Dietary Guidelines for Chinese Residents*, whereas 84.95% expressed willingness to adopt dietary guidance from dieticians. Regarding self-perceived dietary habits, 65.53% considered their eating patterns “healthy”. 63.16% reported not using nutritional supplements. Avoidance of specific seafood items was reported by 16.31% for marine fish, 16.92% for shrimp, and 16.92% for crab. Nearly all patients engaged solely in light physical activity 97.09%. Appetite status was generally reported as “good” (61.95%), with disease-related symptoms (14.09%) and constipation (12.71%) identified as primary appetite-limiting factors.

### 3.5. Subjective Evaluation and Objective Situation

#### 3.5.1. Physician Evaluations and Inpatient Perceptions

The chi-square test results in Figure 2 demonstrated that physician evaluations and patient perceptions showed similar distributions across the five rating categories, with no statistically significant difference in scoring patterns between the two groups (*χ*^2^ = 1.465, *p* > 0.05).

#### 3.5.2. Dietary Intake and Inpatient Perceptions

Figure 3A,B revealed that patients with “very good” self-perceptions exhibited the highest energy and protein intake levels, with only 34.88% and 13.95% below 60% of energy and protein targets. The “good” category showed moderate intake, though 49.26% failed to meet 60% of energy needs. All “bad” and “very bad” patients fell below 60% energy targets, with severely low protein intake (55.26% and 45.06%, respectively). The ordinal logistic regression analysis in Table 2 demonstrates a statistically significant discrepancy between patients’ self-perceived nutritional status and actual achievement rates of both energy and protein intake targets (*p* < 0.05). The multivariable logistic regression analysis was conducted using inpatient self-perception as the dependent variable and both energy and protein intake achievement levels (fully achieved ≥ 100% of targets, partially achieved 60–99% of targets, and not achieved < 60% of targets) as independent variables, with coding and value assignments detailed in Table 3. The analysis demonstrated that the “partially achieved” protein intake group significantly influenced patient perceptions, with an *OR* of 4.044 (95% CI: 1.370–11.936, *p* < 0.05). Significant discrepancies were observed between self-perceptions and both energy and protein achievement levels (*p* < 0.05), except for the “not achieved” protein group (*p* > 0.05), as shown in Table 4.

#### 3.5.3. Dietary Intake and Physician Evaluations

Figure 3C,D demonstrated that even in the physician-rated “very good” group, 40.00% of patients failed to achieve 60% of their target energy intake. The “good” group showed moderate intake, with 46.81% falling below the 60% energy threshold. Notably, the “general” group maintained a relatively high protein intake (82.66%) despite lower energy achievement. All patients in the “very bad” group and 66.67% in the “bad” category did not meet their energy targets, with protein intake declining to 46.27% and 71.91% of requirements, respectively. The ordinal logistic regression analysis presented in Table 5 reveals statistically significant discrepancies between physician evaluations of nutritional status and objective achievement rates for both energy and protein intake targets (*p* < 0.05). The multivariable logistic regression analysis was conducted using physician evaluations as the dependent variable and both energy and protein intake achievement levels (fully achieved ≥ 100% of targets, partially achieved 60–99% of targets, and not achieved < 60% of targets) as independent variables, with coding and value assignments detailed in Table 3. The analysis revealed that both the “partially achieved” (*OR* = 3.247, 95% CI: 1.170–9.013) and “not achieved” (*OR* = 3.809, 95% CI: 1.092–13.283) protein intake groups significantly influenced physician evaluations (*p* < 0.05) However, physician evaluations were significantly inconsistent with all achievement levels for both energy and protein intake (*p* < 0.05), as presented in Table 6.

### 3.6. Blood Parameters

#### 3.6.1. Blood Parameters and NRS 2002 Scores

The correlation analysis in Table 7 reveals significant associations between NRS 2002 nutritional risk scores and specific blood parameters, with renal function, glucose metabolism, inflammation, and nutritional parameters demonstrating the strongest relationships. The analysis identified statistically significant negative correlations between NRS 2002 scores and eGFR (*r* = −0.353, *p* < 0.05), ALB (*r* = −0.211, *p* < 0.05), Hb (*r* = −0.197, *p* < 0.05), and IP (*r* = −0.182, *p* < 0.05), while positive correlations emerged with HbA1c (*r* = 0.357, *p* < 0.05), CRP (*r* = 0.446, *p* < 0.05), BUN (*r* = 0.150, *p* < 0.05), and Ca (*r* = −0.150, *p* < 0.05). Notably, the correlation analysis revealed no statistically significant associations between NRS 2002 scores and various blood parameters, including blood lipids (TC, TG, HDL, LDL), liver function (ALT, AST), or electrolytes (Na, K) (*p* > 0.05), with correlation coefficients ranging from *r* = −0.116 to *r* = 0.073.

#### 3.6.2. Blood Parameters and GLIM Diagnosis

The ordinal logistic regression analysis demonstrates significant progression thresholds across malnutrition categories in Table 8. Compared to the severe malnutrition reference, both moderate malnutrition and non-malnutrition status showed significant separation in ordinal positioning (*p* < 0.05). Several blood parameters showed significant associations with malnutrition severity: Na (*OR* = 2.155, 95% CI: 1.127–4.121) and K (*OR* = 257.752, 95% CI: 2.428–2.74 × 10^4^) levels demonstrated maintained directional agreement with malnutrition severity grading, as did CRP (*OR* = 1.056, 95% CI: 1.001–1.116), indicating potential roles of electrolyte imbalance and systemic inflammation in malnutrition progression (*p* < 0.05). In contrast, conventional nutritional parameters including ALB and Hb, blood lipids (TC and LDL), liver function (ALT), renal function (eGFR), and glucose metabolism parameters (HbA1c) showed no statistically significant associations (*p* > 0.05).

## 4. Discussion

Due to the limitations of the cardiology ward setting, this study did not incorporate muscle mass as a criterion for GLIM-defined malnutrition diagnosis. This decision was further supported by Cederholm et al.’s findings that no consensus has been established regarding the optimal method for measuring and defining reduced muscle mass, particularly in clinical settings [41]. Furthermore, many assessment techniques, including bioelectrical impedance analysis (BIA), ultrasound, computed tomography (CT), and magnetic resonance imaging (MRI), remain unavailable in most nutritional evaluation environments worldwide [41]. Both the American Society for Parenteral and Enteral Nutrition (ASPEN) and ESPEN, along with the Parenteral and Enteral Nutrition Society of Asia (PENSA), recognized BMI and body composition as more critical variables among the phenotypic criteria for malnutrition. BIA, calf circumference, and mid-arm muscle circumference lacked validated clinical cut-off values for muscle mass and were not scientifically robust. Consequently, this limitation may have introduced some degree of bias in the severity grading of malnutrition diagnoses [42]. Therefore, the number of severe malnutrition cases in this study did not reach the expected sample size. WHO recommended a minimum protein intake of 0.8 g·kg^−1^·d^−1^ [43]. Previous studies suggested that while the target protein intake should be 0.8 g·kg^−1^·d^−1^, initial intake could start at 0.6 g·kg^−1^·d^−1^ with gradual increase under renal function monitoring to prevent excessive renal burden [44,45]. However, for chronic kidney disease (CKD) patients, the appropriate daily protein intake varied according to kidney disease stages, and both insufficient (<0.8 g·kg^−1^·d^−1^) or excessive (>1.4 g·kg^−1^·d^−1^) protein intake were associated with increased mortality risk [46]. Therefore, pending further evidence, this study determined target protein intake based on individual patients’ eGFR levels.

The present study found that CVD inpatients had a mean NRS 2002 nutritional risk score of 3.01 ± 0.94, with an overall nutritional risk prevalence of 71.84%. Previous studies demonstrated that the prevalence of nutritional risk among CVD inpatients was 18.51% and 46.3% [47,48]. Lara Hersberger et al. identified through NRS 2002 screening that all hospitalized chronic heart failure patients were at nutritional risk [49]. Other studies reported nutritional risk prevalence rates of 64.41% among hospitalized tuberculosis (TB) patients and 62.3% in Crohn’s disease inpatients [50,51]. However, the higher prevalence observed in our study may be attributed to differences in disease pathology and the older average age of our study population. Concurrently, studies demonstrated that the NRS 2002 is an effective tool for identifying nutritional issues in patients with CKD, diabetes mellitus (DM), and TB. Early detection and management of malnutrition were shown to improve clinical outcomes in these patient populations [50,52,53,54]. The current study identified a malnutrition diagnosis rate of 21.4% among hospitalized CVD patients, which was notably lower than the 40% prevalence reported by Zhou et al. for heart failure inpatients [55]. This discrepancy may be attributed to our study’s exclusion of muscle mass assessment as a diagnostic criterion [55]. Subsequent analyses showed that GLIM-defined malnutrition was significantly associated with both physical and functional impairment [22] and increased all-cause mortality, particularly when coexisting with renal dysfunction [24]. Previous studies demonstrated significant associations between nutritional status and both dietary intake and health-related outcomes (including length of hospitalization, rehospitalization rates, and in-hospital mortality). Malnutrition and consumption of ≤25% of provided meals were each shown to triple the odds of in-hospital mortality within 90 days of admission among obese patients [56,57]. Additionally, nutritional intervention was found to prevent or reduce malnutrition in cancer patients [58]. Specific modalities such as oral nutritional supplements (ONS) and enteral nutrition (EN) demonstrated potential benefits in improving quality of life, morbidity, and mortality among gastric cancer patients [59]. Notably, for malnourished elderly patients with pneumonia, nutritional intervention significantly improved nutritional status and reduced hospital readmission rates [60]. These collective findings underscored the clinical imperative for prioritizing nutritional intervention in GLIM-defined malnutrition cases, particularly emphasizing early intervention for moderately malnourished patients, while highlighting the need for intensive therapeutic management in severe malnutrition cases despite their lower prevalence. The study revealed significant deficiencies in energy and protein intake, with actual energy intake reaching only 63.09% of the target value and protein intake merely 74.98% (*p* < 0.05). This finding was consistent with the results reported by Schuetz et al., who demonstrated that among patients with nutritional risk (NRS 2002 ≥ 3) and an anticipated hospital stay exceeding four days, the achievement rates for energy and protein intake were 79% and 76%, respectively [61]. Consequently, the implementation of integrated digital tools—particularly mobile applications and dietary image recognition devices—was appropriately promoted to facilitate objective quantification of nutritional intake [62,63]. Standardizing hospital meal provision protocols and enhancing communication between dietitians and canteen staff were also critically important.

The dietary nutrition and cognitive behavior questionnaire survey revealed that 32.5% of patients demonstrated confusion regarding scientific dietary practices, while 82.0% had never been exposed to the *Dietary Guidelines for Chinese Residents*, indicating significant gaps in nutritional knowledge dissemination and deficiencies in the nutrition education system. Nutritional information sources were fragmented, primarily consisting of “network” and “own ideas”, with 30.1% of patients encountering contradictory information. The utilization rate of professional channels (physicians and dietitians) remained low, highlighting the need to standardize in-hospital nutrition education pathways. Although 84.9% of patients expressed willingness to receive dietary guidance from nutrition professionals, only 56.3% had received any form of nutrition education, demonstrating an insufficient supply of clinical nutrition support and education. The present study found low supplemental use rates among inpatients (calcium/multivitamins: 8.7% each; fish oil: 5.7%; vitamin D: 0%). Chinese and American studies found that multivitamin supplements were the most commonly used dietary supplements across all age groups, and supplement usage increased with advancing age [64,65]. This underscored the necessity for evidence-based supplementation protocols developed in accordance with clinical guidelines. Widespread dietary misconceptions were observed, with 91.2% of patients inaccurately self-assessing their diets as “healthy” or “very healthy”. These findings aligned with Cong et al.’s research on inpatients in the oncology department, which similarly identified prevalent nutritional knowledge deficits and suboptimal dietary behaviors [66]. Appetite was predominantly inhibited by disease-related symptoms (33.7% cumulative) and digestive issues (26.1% cumulative for constipation/bloating/dyspepsia). Notably, 97.1% of patients engaged only in light physical activity, potentially exacerbating the vicious cycle of appetite suppression and inadequate nutritional intake. These results may be attributed to the generally older age and lower education level of patients, who typically lacked nutrition knowledge unless they or their relatives worked in related fields. Several studies demonstrated that intensive nutrition education significantly improved nutritional status and quality of life in gastrectomy patients [67]. Evidence showed nutrition education effectively enhanced adherence to protein intake recommendations among CKD patients while reducing mortality and morbidity rates in end-stage renal disease (ESRD) populations [68,69]. Concurrently, nutrition education played a crucial role in improving glycemic control for DM patients [70]. These results collectively advocate for standardized nutrition education programs, community-based interventions, and enhanced public awareness campaigns, including distribution of educational materials and establishment of official nutrition information platforms, to systematically improve population-wide nutritional literacy.

The chi-square test revealed no significant differences between physicians and inpatients in their assessments of dietary status, indicating a high degree of consistency in evaluation distributions between the two groups. The ordinal logistic regression analysis revealed significant discrepancies between both patient self-perceptions and physician evaluations with actual achievement rates of energy and protein intake targets (*p* < 0.05). Multivariable logistic regression analysis demonstrated that both patient self-perceptions and physician evaluations showed greater attention to protein intake while potentially neglecting energy intake. These findings highlight substantial disparities between subjective evaluations and objective dietary intake measurements. Importantly, the results indicate that neither physicians’ clinical judgments nor patients’ self-perceptions reliably predict true nutritional status. The study revealed that while over half of urban diabetes patients in China had received nutrition education, their nutritional knowledge and dietary practices remained suboptimal [71]. Elderly Chinese populations demonstrated particularly low levels of dietary knowledge, highlighting the need for targeted nutrition education for older patients [72]. Individuals with lower education levels, rural or western China residency, unfamiliarity with the *Dietary Guidelines for Chinese Residents*, and passive approaches to nutrition information-seeking were significantly less likely to possess adequate dietary literacy [73]. Previous studies indicated inadequate nutrition knowledge among general practitioners, with inconsistent confidence in nutritional counseling skills [74]; 65.2% of physicians (including internists and surgeons) displayed insufficient clinical nutrition knowledge [75]. These findings underscore the necessity for enhanced nutrition education to positively influence patient care [76]. Gyeongsil & Seung-Won highlighted the critical need for continuous learning and practical training among clinicians [77]. Additionally, the integration of nutrition courses into medical school curricula was strongly recommended as a critically important initiative to better support future clinicians [78]. However, dietitians’ work was frequently constrained by competing clinical priorities, emphasizing the importance of improving hospital culture and environments through sustainable interventions [79]. In conclusion, strengthened collaboration is needed between dietitians and physicians, along with the integration of clinical dietitians into multidisciplinary teams. This included implementing regular multidisciplinary consultations and establishing concurrent online/offline systems for nutritional monitoring and assessment to achieve optimal nutritional management of hospitalized cardiovascular patients [80,81,82,83].

This study included these blood parameters because elevated LDL, TG, and CRP levels were strongly associated with increased cardiovascular risk, while impaired renal function and poor glycemic control represented common comorbidities in CVD patients [84,85]. Additionally, given malnutrition’s significant negative impact on patient recovery and prognosis, we measured nutritional parameters, including ALB and Hb, to comprehensively assess nutritional status [22,86]. Correlation analysis between blood parameters and NRS 2002 scores demonstrated significant positive associations for both CRP and HbA1c (*p* < 0.05), consistent with previous findings by Nienaber-Rousseau et al., Mottalib et al., and Pourhassan et al., suggesting that inflammatory status and glucose metabolism may contribute to nutritional risk [87,88,89]. Furthermore, studies demonstrated that cancer patients with elevated CRP concentrations exhibited poorer nutritional status [90]. Similarly, in hospitalized patients with acute exacerbations of chronic obstructive pulmonary disease (AECOPD), CRP levels showed significant correlations with nutritional risk [91]. Conversely, eGFR, ALB, and Hb showed significant negative correlations with NRS 2002 scores, potentially serving as predictive biomarkers for nutritional risk, in agreement with studies by Bargetzi et al., Miličević et al., and Zhou et al. [92,93,94]. No significant associations were observed for TC, LDL, ALT, or AST (*p* > 0.05), which contrasted with Guligowska et al.’s findings but partially aligned with Oh et al.’s results showing only ALT was associated, indicating limited utility of these parameters for nutritional risk assessment [95,96]. Ordinal logistic regression analysis of blood parameters against GLIM criteria revealed significant impacts of Na and K on malnutrition grading, supporting Hirose et al.’s conclusions but differing from Oguri et al.’s results, possibly due to age-related metabolic variations [97,98]. These collective findings established CRP as a valid inflammatory marker for GLIM-diagnosed malnutrition [99,100]. The results highlighted the clinical importance of early nutritional risk screening upon hospital admission, incorporating key blood parameters (Hb and ALB). For patients with identified nutritional risk or confirmed malnutrition, prompt nutritional intervention was recommended to reduce severe complications and mortality while improving clinical outcomes. Comprehensive monitoring integrating both nutritional status and renal function emerged as essential in CVD management, with eGFR serving as a critical parameter for developing individualized nutritional strategies and dietary plans.

This study had limitations due to its restriction to a single hospital site and predominantly elderly patient population. Consequently, the findings may not be generalizable to younger cardiovascular patients or other clinical settings, leaving the external validity of the results unresolved, though partial generalizability to other tertiary hospitals in China remains possible. Nevertheless, the findings offer a novel perspective for healthcare providers to better comprehend the actual nutritional status of hospitalized patients, thereby establishing a foundation for future multidimensional and more in-depth investigations. Conducting longitudinal studies, such as comparing clinical outcomes between nutrition intervention and control groups or further exploring the significance of nutritional interventions, could improve the representativeness of the research outcomes.

## 5. Conclusions and Prospects

In conclusion, this investigation of hospitalized patients in the cardiology department demonstrated that the surveyed patients exhibited deficient dietary knowledge, low rates of proper nutritional practices, and widespread dietary misconceptions. Both physician evaluations and inpatient self-perceptions proved inadequate for accurately reflecting the actual nutritional intake status among CVD patients, indicating that subjective evaluations failed to represent true nutritional status. These findings demonstrated that relying solely on physician and patient oral-reported dietary assessments inadequately reflected patients’ true nutritional status. Consequently, comprehensive nutritional evaluation was essential for accurate clinical assessment, while intensified nutrition education initiatives targeting both the general population and healthcare professionals were required to improve hospital dietary environments and infrastructure, thereby enhancing societal nutritional literacy. We strongly advocate for enhanced collaboration between clinicians and dietitians to scientifically evaluate patients’ nutritional status and reduce the incidence of malnutrition among cardiovascular disease patients. Improving physicians’ nutrition education is equally critical, particularly in mastering NRS 2002 nutritional risk screening and 24 h dietary recall methods for hospitalized patients. When providing nutritional education, patients should receive accurate, consistent, and authoritative dietary guidance to optimize their care. There is a need to develop more clinically practical methods for muscle mass measurement. These evidence-based observations provided actionable insights for developing targeted nutritional interventions, offering clinical implications for healthcare practitioners across China, Asia, and global settings.

## Figures and Tables

**Figure 1 nutrients-17-02624-f001:**
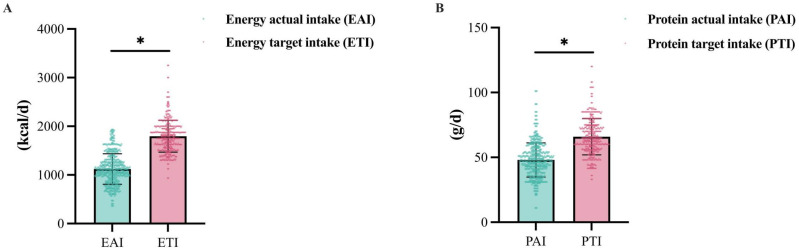
Energy and protein intake in cardiology inpatients: (**A**) energy intake comparison, (**B**) protein intake comparison. EAI, energy actual intake; ETI, energy target intake; PAI, protein actual intake; PTI, protein target intake. *, *p* < 0.05 (two-tailed). There were statistically significant differences between actual and target intakes for both energy and protein.

**Figure 2 nutrients-17-02624-f002:**
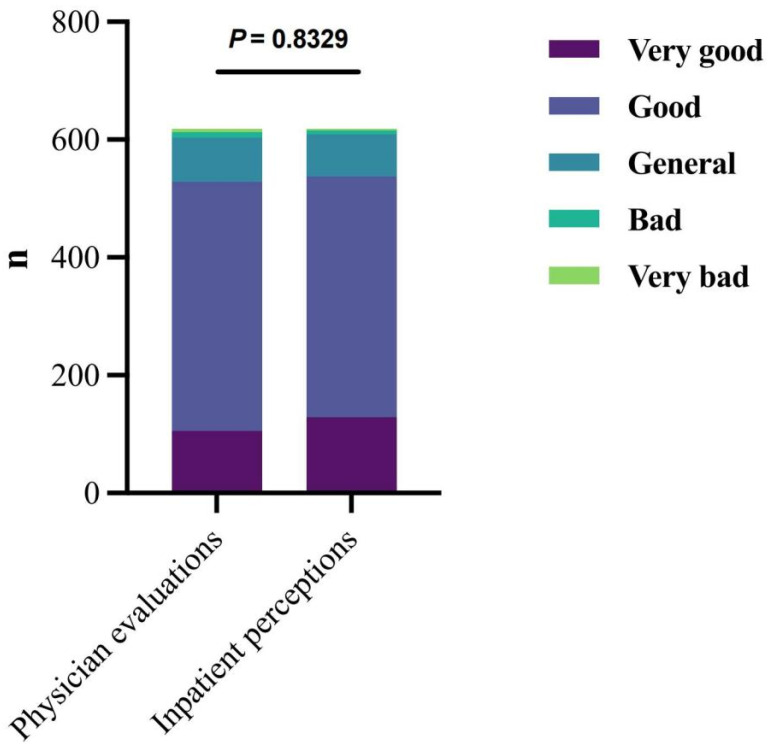
Cardiology inpatient perceptions and physician evaluations of dietary status. Both physicians and inpatients used five rating levels (very good, good, general, bad, very bad), which were represented by five distinct colors. No significant difference between physician evaluations and patient perceptions.

**Figure 3 nutrients-17-02624-f003:**
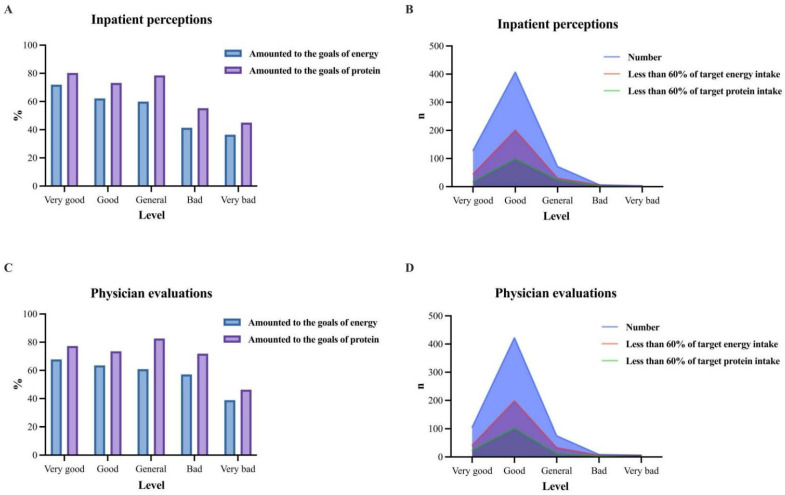
Comparison between subjective evaluations and objective dietary intake: (**A**) energy/protein attainment rates by patient-perceived levels, (**B**) energy/protein attainment rates by physician-evaluated levels, (**C**) patients with energy intake <60% target, and (**D**) patients with protein intake <60% target. The five assessment levels for both physicians and inpatients were as follows: very good, good, general, bad, and very bad.

**Table 1 nutrients-17-02624-t001:** Target intake of energy and protein.

Classification	Patient Category		Recommendation
Target energy intake	Bedridden patients		22 kcal·kg^−1^·d^−1^
	Ambulatory patients		25 kcal·kg^−1^·d^−1^
Target protein intake	General patients		1.0 g·kg^−1^·d^−1^
	CKD patients	eGFR < 15 mL·min^−1^·(1.73 m^2^)^−1^ (dialysis-dependent)	1.0 g·kg^−1^·d^−1^
		eGFR < 30 mL·min^−1^·(1.73 m^2^)^−1^ (non-dialysis)	0.6 g·kg^−1^·d^−1^
		eGFR 30–89 mL·min^−1^·(1.73 m^2^)^−1^	0.8 g·kg^−1^·d^−1^

Footnote: CKD, chronic kidney disease; eGFR, estimated glomerular filtration rate.

**Table 2 nutrients-17-02624-t002:** Ordinal logistic regression analysis of cardiology inpatient perceptions and dietary intake.

Groups	*β*	*SE*	Wald *χ*^2^	*p*	*OR*	95% CI
**Inpatient Perceptions**						
Very good ^a^	-	-	-	-	-	-
Good	2.650	0.575	21.234	0.000	14.157	4.586–43.706
General	−0.738	0.542	1.852	0.174	0.478	0.165–1.384
Bad	−2.850	0.709	16.160	0.000	0.058	0.014–0.232
Very bad	−3.555	0.867	16.826	0.000	0.029	0.005–0.156
**Dietary Intake**						
Achievement rate of target energy intake	0.041	0.012	12.307	0.000	1.042	1.018–1.066
Achievement rate of target protein intake	−0.019	0.009	4.016	0.045	0.982	0.964–1.000

Footnote: ^a^ was the reference group.

**Table 3 nutrients-17-02624-t003:** Coding and value assignment for multivariable logistic regression.

Code	Variable	Assignment
Y_1_	Inpatient perceptions	Very good = 5; Good = 4; General = 3; Bad = 2; Very bad = 1
Y_2_	Physician evaluations	Very good = 5; Good = 4; General = 3; Bad = 2; Very bad = 1
X_1_	Energy intake achievement levels	Fully achieved = 3; Partially achieved = 2; Not achieved = 1
X_2_	Protein intake achievement levels	Fully achieved = 3; Partially achieved = 2; Not achieved = 1

**Table 4 nutrients-17-02624-t004:** Multivariate logistic regression analysis of cardiology inpatient perceptions and dietary intake achievement levels.

Factor	*β*	*SE*	Wald *χ*^2^	*p*	*OR*	95% CI
**Inpatient Perceptions**						
Very good ^a^	-	-	-	-	-	-
Good	0.084	0.572	0.022	0.883	1.088	0.355–3.337
General	−3.356	0.616	29.668	0.000	0.035	0.010–0.117
Bad	−5.482	0.772	50.470	0.000	0.004	0.001–0.019
Very bad	−6.191	0.920	45.326	0.000	0.002	0.000–0.012
**Energy Intake Achievement Degree**						
Fully achieved ^a^	-	-	-	-	-	-
Partially achieved	−2.627	0.739	12.633	0.000	0.072	0.017–0.308
Not achieved	−2.571	0.772	11.079	0.001	0.076	0.017–0.348
**Protein Intake Achievement Degree**						
Fully achieved ^a^	-	-	-	-	-	-
Partially achieved	1.397	0.552	6.400	0.011	4.044	1.370–11.936
Not achieved	0.610	0.640	0.908	0.341	1.841	0.525–6.457

Footnote: ^a^ was the reference group.

**Table 5 nutrients-17-02624-t005:** Ordinal logistic regression analysis of physician evaluations and dietary intake.

Groups	*β*	*SE*	Wald *χ*^2^	*p*	*OR*	95% CI
**Physician evaluations**						
Very good ^a^	-	-	-	-	-	-
Good	2.225	0.574	15.032	0.000	9.250	3.004–28.482
General	−1.263	0.554	5.199	0.023	0.283	0.096–0.838
Bad	−3.230	0.691	21.846	0.000	0.040	0.010–0.153
Very bad	−4.162	0.881	22.335	0.000	0.016	0.003–0.088
**Dietary intake**						
Achievement rate of target energy intake	0.034	0.012	8.887	0.003	1.035	1.012–1.059
Achievement rate of target protein intake	−0.022	0.009	5.356	0.021	0.979	0.961–0.997

Footnote: ^a^ was the reference group.

**Table 6 nutrients-17-02624-t006:** Multivariate logistic regression analysis of physician evaluations and dietary intake achievement levels.

Factor	*β*	*SE*	Wald *χ*^2^	*p*	*OR*	95% CI
**Physician Evaluations**						
Very good ^a^	-	-	-	-	-	-
Good	1.087	0.567	3.669	0.055	2.965	0.975–9.016
General	−2.391	0.592	16.297	0.000	0.092	0.029–0.292
Bad	−4.368	0.718	37.062	0.000	0.013	0.003–0.052
Very bad	−5.303	0.903	34.522	0.000	0.005	0.001–0.029
**Energy Intake Achievement Degree**						
Fully achieved ^a^	-	-	-	-	-	-
Partially achieved	−1.584	0.660	5.759	0.016	0.205	0.056–0.748
Not achieved	−1.956	0.706	7.680	0.006	0.141	0.035–0.564
**Protein intake Achievement Degree**						
Fully achieved ^a^	-	-	-	-	-	-
Partially achieved	1.178	0.521	5.112	0.024	3.247	1.170–9.013
Not achieved	1.337	0.637	4.404	0.036	3.809	1.092–13.283

Footnote: ^a^ was the reference group.

**Table 7 nutrients-17-02624-t007:** Correlation analysis between blood parameters and NRS 2002.

Classification	Parameters	Results	Normal Reference Range	*r*	*p*
Blood lipids	TC (mmol/L)	4.08 ± 1.16	2.90–6.20	−0.116	0.124
	TG (mmol/L)	1.58 ± 1.35	0.45–1.70	−0.085	0.241
	HDL (mmol/L)	1.07 ± 0.32	1.03–1.55	−0.002	0.980
	LDL (mmol/L)	2.23 ± 0.86	1.90–4.10	−0.110	0.130
Liver function	ALT (U/L)	22.74 ± 24.84	Male: 9–50, female: 7–40	0.021	0.771
	AST (U/L)	22.82 ± 15.72	Male: 15–40, female: 13–35	0.073	0.300
Renal function	BUN (mmol/L)	6.11 ± 2.61	2.8–7.2	0.150	0.031 *
	Cr (µmol/L)	77.86 ± 39.58	Male: 59–104, female: 45–84	0.133	0.057
	eGFR (mL·min^−1^·(1.73 m^2^)^−1^)	88.43 ± 20.27	-	−0.353	<0.001 *
Glucose metabolism	Glu (mmol/L)	5.85 ± 2.86	3.3–6.1	0.091	0.192
	HbA1c (%)	7.14 ± 4.41	4.0–6.0	0.357	<0.001 *
Inflammation	CRP (mg/L)	8.98 ± 26.03	0–10	0.446	<0.001 *
Nutrition	ALB (g/L)	41.29 ± 4.31	Male: 40–50, female: 40–55	−0.211	0.002 *
	Hb (g/L)	133.78 ± 18.97	Male: 130–175, female: 115–150	−0.197	0.005 *
Electrolyte	Na (mmol/L)	140.72 ± 2.76	137–147	−0.067	0.340
	K (mmol/L)	3.99 ± 0.40	3.5–5.3	0.006	0.933
	Ca (mmol/L)	2.28 ± 0.13	2.2–2.65	−0.150	0.031 *
	IP (mmol/L)	1.21 ± 0.22	0.80–1.45	−0.182	0.009 *
Metabolism	UA (µmol/L)	335.89 ± 98.26	Male: 208–428, female: 155–357	0.085	0.227

Footnote: * *p* < 0.05 (two-tailed); TC, total cholesterol; TG, triglycerides; HDL, high-density lipoprotein; LDL, low-density lipoprotein; ALT, alanine aminotransferase; AST, aspartate aminotransferase; BUN, blood urea nitrogen; Cr, creatinine; UA, uric acid; GFR, glomerular filtration rate; Glu, glucose; HbA1c, glycated hemoglobin; CRP, C-reactive protein; ALB, albumin; Hb, hemoglobin; Na, sodium; K, potassium; Ca, calcium; IP, inorganic phosphate.

**Table 8 nutrients-17-02624-t008:** Ordinal logistic regression analysis of blood parameters and GLIM diagnosis.

Classification	Parameters	*β*	*SE*	Wald *χ*^2^	*p*	*OR*	95% CI
GLIM diagnosis	Severe malnutrition ^a^	-	-	-	-	-	-
	Moderate malnutrition	113.438	49.407	5.272	0.022	1.84 × 10^49^	1.62 × 10^7^–2.09 × 10^91^
	Non-malnutrition	124.042	51.484	5.805	0.016	7.43 × 10^53^	1.11 × 10^10^–4.95 × 10^97^
Blood lipids	TC	1.043	4.080	0.065	0.798	2.837	0.001–8.43 × 10^3^
	TG	0.673	0.929	0.524	0.469	1.960	0.317–12.110
	HDL	5.844	4.284	1.861	0.173	3.45 × 10^02^	0.078–1.53 × 10^6^
	LDL	−2.312	4.888	0.224	0.636	0.099	6.84 × 10^−6^–1.43 × 10^3^
Liver function	ALT	0.033	0.047	0.500	0.479	1.0336	0.943–1.134
	AST	0.036	0.025	1.979	0.159	1.0367	0.986–1.089
Renal function	BUN	0.559	0.377	2.199	0.138	1.749	0.835–3.666
	Cr	−0.072	0.051	1.978	0.160	0.931	0.842−1.028
	eGFR	0.038	0.071	0.278	0.598	1.039	0.903–1.193
Glucose metabolism	Glu	−0.428	0.275	2.425	0.119	0.652	0.381–1.117
	HbA1c	0.029	0.178	0.026	0.872	1.029	0.726–1.458
Inflammation	CRP	0.055	0.028	3.918	0.048	1.056	1.001–1.116
Nutrition	ALB	0.077	0.163	0.220	0.639	1.080	0.784–1.487
	Hb	−0.022	0.038	0.320	0.571	0.978	0.908–1.054
Electrolyte	Na	0.768	0.331	5.395	0.020	2.155	1.127–4.121
	K	5.552	2.380	5.440	0.020	2.58 × 10^2^	2.428–2.74 × 10^4^
	Ca	−8.665	7.168	1.461	0.227	0.000	1.37 × 10^−10^–2.18 × 10^2^
	IP	−1.981	2.659	0.555	0.456	0.138	0.001–25.305
Metabolism	UA	0.007	0.007	0.927	0.336	1.007	0.993–1.021

Footnote: ^a^ was the reference group.

## Data Availability

The original contributions presented in this study are included in the article/Appendix A. Further inquiries can be directed to the corresponding author.

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
