# Peer review of "Mismatch Between Perceived and Actual Dietary Nutrition in Hospitalized Cardiovascular Patients and Clinicians: A Cross-Sectional Assessment and Recommendations for Improvement"

_nutrients, 2025, doi:10.3390/nu17162624_

Round 1

Reviewer 1 Report

Comments and Suggestions for Authors

Was the random sampling method truly random, and how were potential selection biases controlled for in patient recruitment?
How do the authors justify drawing conclusions about clinical practice and intervention needs based on a cross-sectional design without outcome data?
Why was muscle mass, a core phenotypic criterion of GLIM, excluded rather than estimated via proxy measures (e.g., calf circumference, handgrip strength)?

What measures were taken to validate or compare reported intake to actual food provisions given the known limitations of the 24-hour dietary recall?
Was the self-designed questionnaire validated prior to implementation? If not, what do the authors say about its lack of internal consistency (i.e., Cronbach's alpha ~0.6-0.7)?

Was the sample size (n=618) adequately powered to detect differences in multivariable models, especially for severe malnutrition (only 18 cases)?
While the authors find no significant differences in the distribution of physician vs. patient evaluations, how do they reconcile this with the large discrepancies in actual intake data?

How clinically meaningful are correlations between NRS 2002 scores and biochemical markers (e.g., CRP, eGFR), based on the modest r-values (e.g., −0.353 to +0.446)? Were comorbidities (e.g., diabetes, CKD) statistically controlled to analyze associations with malnutrition or gaps in intake? To what extent can the results be generalized beyond a single tertiary care hospital in China? Could the diet culture or the infrastructure of the hospital affect the conclusions? Does this study provide evidence that better nutritional education or better dietary intake would actually improve patient outcomes (e.g., mortality, complications, rehospitalization)?
How feasible is it to expect physicians to improve dietary counseling skills without systemic changes, such as integrating dietitians into the cardiology care team?
The conclusion advocates for real-time dietary image recognition. Has this been tested in similar populations, and is it realistic in low-resource settings?
What specific intervention trials do the authors propose to address the gaps found in subjective nutritional assessment and actual intake?
Would a structured, dietitian-led inpatient educational program improve patient knowledge or intake more effectively than current ad-hoc counseling?

Reviewer 2 Report

Comments and Suggestions for Authors

I read with interest the manuscript “Association Between Dietary Nutrition Awareness of Hospitalized Cardiovascular Patients and Clinicians: Current Status and Improvement Suggestions”.

This manuscript presents an important and timely cross-sectional study examining the alignment between subjective and objective nutritional assessments among hospitalized cardiovascular disease (CVD) patients, along with their clinicians. The research is novel in design, methodologically solid, and clinically meaningful. It underscores the existing cognitive gap in nutritional awareness and provides evidence supporting enhanced collaboration between clinicians and dietitians.

Major concern

  1. The objective of evaluating the consistency between subjective dietary evaluations (by patients and physicians) and actual intake is clear. However, the hypothesis is not explicitly stated. It would be beneficial to articulate this early in the introduction, for instance: “We hypothesized that subjective perceptions of nutritional status would not align with actual dietary intake and objective nutritional indicators”.
  2. The study's core strength lies in showing that subjective evaluations (from both physicians and patients) poorly reflect true energy and protein intake. This is a valuable clinical insight. However, the discussion could be further strengthened by providing specific examples of how misalignment may affect clinical outcomes (e.g., missed intervention opportunities, undernutrition going unnoticed).
  3. Due to the lack of measurement tools, the GLIM malnutrition diagnosis omitted muscle mass assessment. While this is acknowledged, it limits the diagnostic accuracy. A more in-depth discussion of the potential underestimation of malnutrition prevalence should be included.
  4. Although the sample is adequate, it’s restricted to one hospital and primarily includes elderly patients. The external validity of the findings (e.g., application to younger CVD populations or other clinical settings) is not addressed. Please add a short paragraph on this in the limitations.
  5. The conclusion argues for enhanced clinician-dietitian collaboration and better nutrition education, but no clear roadmap or strategic framework is offered. Consider suggesting specific interventions, such as routine use of dietitian-administered 24-h recalls, in-hospital nutrition training for medical staff, or integration of digital tools for intake monitoring.

Minor comments:

  1. Abstract: Clarify statistical results (e.g., P-values) with context: what do they mean for practice?
  2. Some acronyms (e.g., NRS 2002, GLIM, eGFR) are defined only once or inconsistently used. Make sure each acronym is explained upon first mention in both the abstract and main text.

Reviewer 3 Report

Comments and Suggestions for Authors

Dear Authors,

Thank you for your manuscript. Please see my comments below.

Study title. Based on the descriptive nature of the study, the observed mismatch between subjective evaluations and actual dietary intake, and the inclusion of both patients and clinicians, I recommend revising the title for better clarity and accuracy. The current title implies an analytical association, which is not supported by the statistics. A more appropriate title could be:
"Mismatch Between Perceived and Actual Dietary Nutrition in Hospitalized Cardiovascular Patients and Clinicians: A Cross-Sectional Assessment and Recommendations for Improvement." This better reflects the descriptive comparison presented in the paper. Since no statistical comparisons or associations between inpatients and clinicians are performed, the current phrasing is misleading.

The abstract should provide information on the sample size, mean age (SD) and distribution in gender groups.

Introduction. Currently starts with global cardiovascular burden and generic risk factors (e.g. smoking, hypertension). This distracts from the study's focus. Also, the Introduction lacks a clear definition of malnutrition, particularly in the CVD inpatient context, omits discussion of why patients and clinicians may misjudge nutritional status and makes no explicit statement of the study’s novelty, e.g. evaluating the alignment of perception vs. objective nutrition intake.

Methods. No description of the study participants (sociodemographics, health condition) and recruitment criteria is presented.

Statistics. From section 2.7 (Statistics), it remains unclear what the calculated sample size is. This information is presented only later in the baseline characteristics.

Results. The main text is overloaded with tables (11) and figures (3). Tables 1–4 are purely descriptive, not comparative or inferential. This makes them suitable for Supplementary Material.

Reviewer 4 Report

Comments and Suggestions for Authors

The topic of the manuscript is clinically relevant, addressing a critical issue—malnutrition and nutritional assessment among hospitalized cardiovascular disease (CVD) patients. The study examines both subjective and objective nutritional assessments, linking them to intake data and biochemical markers, which offers a well-rounded view.

The abstract is densely packed, especially in the Objective and Methods sections, reducing readability.

It’s unclear how many total patients were included in the study. The number diagnosed with malnutrition (132) is mentioned, but the total sample size is missing.

The phrase “actual dietary intake was recorded” lacks detail. How was this measured—food weighing, observation, recall? The method of assessing intake directly affects validity.

The reported test statistic and p-value for agreement between physician and patient evaluations are inconsistent: “χ²=1.465, P<0.05” seems contradictory since such a small chi-square value typically corresponds to a non-significant result. This needs clarification or correction.

More detail is needed on the regression models used. For instance, were confounding variables controlled? What were the effect sizes?

The phrase “subjective evaluations couldn’t accurately reflect actual nutritional status” would be stronger if supported with specific sensitivity/specificity or agreement statistics.

The term “cognitive bias” may be overly broad and potentially misleading. The authors should clarify whether they mean lack of awareness, misperception, or another specific form of bias.

Subjective evaluations “necessitate strengthened monitoring,” but does not specify how this should be operationalized. While this may be addressed in the full paper, a brief indication of recommended interventions or monitoring tools would enhance the conclusion.

Round 2

Reviewer 1 Report

Comments and Suggestions for Authors

The reviewer is very disappointed as a lot of the answers were 100% generated using AI technology.

Author Response

Thanks. We have improved the point-by-point replies. Please see the attachment.

Reviewer 2 Report

Comments and Suggestions for Authors

Well done

Author Response

Thanks!

Reviewer 3 Report

Comments and Suggestions for Authors

Dear Authors,

I appreciate taking into account all my comments. Now the paper seems to be much improved. I have no comments.

All the best.

Author Response

Thank you.

Reviewer 4 Report

Comments and Suggestions for Authors

Having seen the manuscript's modifications, I agree with its publication, as it has been improved.

Author Response

Thanks.

Round 3

Reviewer 1 Report

Comments and Suggestions for Authors

The paper can be accepted in its present form.